# Pregnancy loss and risk of multiple sclerosis and autoimmune neurological disorder: A nationwide cohort study

Anders Pretzmann Mikkelsen[1,2]*, Pia Egerup[2,3,4], Astrid Marie Kolte[2,4], David Westergaard[4,5,6], Henriette Svarre Nielsen[2,3,4], Øjvind Lidegaard[1,2]

1 Department of Gynaecology, Copenhagen University Hospital Rigshospitalet, Copenhagen, Denmark, 2 Department of Clinical Medicine, University of Copenhagen, Copenhagen, Denmark, 3 Department of Obstetrics and Gynaecology, Copenhagen University Hospital Hvidovre, Hvidovre, Denmark, 4 The Recurrent Pregnancy Loss Unit, The Capital Region, Copenhagen University Hospitals Rigshospitalet and Hvidovre, Copenhagen, Denmark, 5 Novo Nordisk Foundation Center for Protein Research, University of Copenhagen, Copenhagen, Denmark, 6 Methods and Analysis, Statistics Denmark, Copenhagen, Denmark

* anders.mikkelsen@regionh.dk

**Data Availability Statement:** The underlying data used in this study is not publicly available and cannot be made publicly available by the authors due to Danish legal restrictions. Researchers can

## Abstract

### Background

The loss of one or more pregnancies before viability (i.e. pregnancy loss or miscarriage), has been linked to an increased risk of diseases later in life such as myocardial infarction and stroke. Recurrent pregnancy loss (i.e. three consecutive pregnancy losses) and multiple sclerosis have both been linked to immunological traits, which could predispose to both occurrences. The objective of the current study was to investigate if pregnancy loss is associated with later autoimmune neurological disease.

### Methods

This register-based cohort study, included the Danish female population age 12 or older between 1977–2017. Women were grouped hierarchically: 0, 1, 2, ≥3 pregnancy losses, primary recurrent pregnancy loss (i.e. not preceded by a delivery), and secondary recurrent pregnancy loss (i.e. preceded by a delivery). The main outcome was multiple sclerosis and additional outcomes were amyotrophic lateral sclerosis, Guillain-Barré syndrome, and myasthenia gravis. Bayesian Poisson regression estimated incidence rate ratios [IRR] and 95% credible intervals [CI] adjusted for year, age, live births, family history of an outcome, and education.

### Results

After 40,380,194 years of follow-up, multiple sclerosis was diagnosed among 7,667 out of 1,513,544 included women (0.5%), median age at diagnosis 34.2 years (IQR 27.4–41.4 years), and median age at symptom onset 31.2 years (IQR 24.8–38.2). The adjusted IRR of multiple sclerosis after 1 pregnancy loss was: 1.03 (95% CI 0.95–1.11), 2 losses: 1.02 (95% CI 0.86–1.20), ≥3 non-consecutive losses: 0.81 (95% CI 0.51–1.24), primary recurrent pregnancy loss: 1.18 (95% CI 0.84–1.60), secondary recurrent pregnancy loss: 1.16 (95%

apply for access to Danish healthcare data at the Danish Health Data Authority (https://sundhedsdatastyrelsen.dk/da/english). Aggregated versions of the original data can be made available upon reasonable request to the corresponding author. We confirm the data located in the paper and its Supporting Information files constitutes the minimal data set.

**Funding:** All funding for the study were provided by grants from The Research Fund of Rigshospitalet, Copenhagen University Hospital [grant number E-22515-01] awarded to APM and ØL. URL: https://www.forskningspuljer-rh.dk/ and Ole Kirks Foundation [no grant number] awarded to HSN. URL: https://www.olekirksfond.dk/). There was no additional external funding received for this study. The funders had no role in study design, data collection and analysis, decision to publish, or preparation of the manuscript.

**Competing interests:** Amended Competing Interests statement: The authors report no conflicts of interest for the current study. For funding of research not relating to the current study, authors report: HSN received research grants from commercial sources: Freya Biosciences ApS, Ferring Pharmaceuticals, BioInnovation Institute, and non-commercial sources: Novo Nordisk Foundation, Ministry of Education, Augustinus Foundation, and Oda and Hans Svenningsens Foundation. For funding of research not relating to the current study DW reports a research grant from non-commercial source: Novo Nordisk Foundation. For work not relating to the current study HSN reports honoraria for lectures (speakers-fee) from commercial sources: Ferring Pharmaceuticals, Merck A/S, Astra Zenica, and Cook Medical. The funders had no role in study design, data collection and analysis, decision to publish, or preparation of the manuscript. No authors report employment, consultancy, patents, or work relating to products in development, or marketed products. This does not alter our adherence to PLOS ONE policies on sharing data and materials.

CI 0.81–1.63), as compared to women with no pregnancy losses. Seven sensitivity analyses and analyses for additional outcomes did not show significantly elevated adjusted risk estimates.

## Conclusions

In this nationwide study, pregnancy loss was not significantly associated with autoimmune neurological disorder.

## Introduction

Multiple sclerosis is the most prevalent demyelinating neurological disease and the incidence highest among women of reproductive age [1]. Although the etiology is unknown, the disease is presumed to be an autoimmune disorder with a hereditary element, were disease development is effected by unknown environmental interactions [2, 3]. Single nucleotide polymorphisms in genes essential for immune regulation; such as interleukin receptor genes, and genes in the human leukocyte antigen (HLA) locus, are associated with multiple sclerosis [4, 5].

During pregnancy, changes in maternal immune response occur in order not to reject the genetically dissimilar fetus [6]. Whether this mechanism could be responsible for the advantageous course of multiple sclerosis during pregnancy is unknown [7]. Pregnancy loss occurs in 10–15% of pregnancies seen in a hospital setting or confirmed by ultrasonography [8, 9]. Although most pregnancy losses are due to fetal genetic defects [10], some pregnancy losses may be due to maternal autoimmune disease and possibly defects in feto-maternal immune interactions [11, 12]. The fetal rate of aneuploidy decreases with increasing number of pregnancy losses [13]. Recurrent pregnancy loss, in Denmark defined as three consecutive pregnancy losses, (although some countries use a definition of two consecutive losses), occurs for 1–2% of women trying to conceive [14, 15]. Recurrent pregnancy loss can be divided into primary recurrent pregnancy loss which is not preceded by a delivery, and secondary recurrent pregnancy loss which is preceded by a delivery. Primary and secondary pregnancy loss are possibly linked to a different spectrum of diseases [16, 17], and have further been associated to specific maternal HLA alleles [18, 19].

The current study investigates the association between multiple and recurrent pregnancy loss and risk of developing autoimmune neurological disorders.

## Methods

### Study design

Register-based historical cohort study.

### Participants

All women born between 1957–1997 and living in Denmark between 1977–2017 were eligible for inclusion in the study cohort and identified in the Danish Civil Registration System [20]. The unique personal identification number provided to each resident enabled linkage to other national registers. Immigrants were eligible for inclusion if immigration occurred before age 20. Persons with an outcome of interest before age 12 were excluded. Follow-up commenced at age 12, immigration, or start of follow-up (January 1, 1977), whichever came last. Censoring occurred due to emigration, death, end of follow-up (December 31, 2017), or an event of

interest, whichever came first. The study was approved by the Danish Health Data Authority. No informed consent is need for register-based studies in Denmark. A complete list of definitions and registers used can be found in the S1 Table.

### Reproductive history

In the Danish Medical Birth Register [21], the number of live births and stillbirths were identified. The lower gestational threshold for stillbirths in this register was 28 gestational weeks before 2004 and 22 gestational weeks after. Therefore, pregnancies ending at earlier gestational ages were identified in the Danish National Patient Register [22]. A pregnancy loss was defined as the registered loss of a pregnancy before viability and included miscarriage, blighted ovum (ie. a pregnancy with a visible gestational sac, but no visible embryo), and missed abortion (ie. a nonviable pregnancy remaining in the uterus). Recurrent pregnancy loss was defined as three consecutive pregnancy losses without a delivery or induced abortion in between or a specific diagnosis of recurrent pregnancy loss. Recurrent pregnancy loss was subdivided by whether the sequence was preceded by a delivery (secondary) or not (primary). As complications of early pregnancy could lead to multiple hospital contacts during clinical care, an algorithm using restriction periods was used (S1 Appendix) to ascertain each pregnancy was only counted once. Therefore, additional pregnancy outcomes were also identified: ectopic pregnancy, molar pregnancy, and induced abortion, to separate individual pregnancies. The latter were identified in the Danish Register of Legally Induced Abortions [23].

### Exposure

Pregnancy loss was the exposure of interest and defined hierarchically with no pregnancy losses at the lowest level and recurrent pregnancy loss at the highest level, as these women were a priori hypothesized to have a higher probability of underlying immune disorder. Women could increase exposure level during follow-up but not revert, and could only contribute risk time in one category at a given time during follow-up (categories: no pregnancy losses, 1 pregnancy loss, 2 pregnancy losses, ≥3 non-consecutive pregnancy losses [not fulfilling criteria for recurrent pregnancy loss], primary recurrent pregnancy loss, and secondary recurrent pregnancy loss).

### Outcome

The primary outcome of interest was an incident diagnosis of multiple sclerosis in the Danish Multiple Sclerosis Register [24]. In case the date of diagnosis or date of symptom onset was given as the calendar year, June 1 was chosen as the date of diagnosis or symptom onset. Kinship information was available since 1968 in the Civil Registration System. As the Danish Multiple Sclerosis Register was only available for women, multiple sclerosis among first degree relatives (mother, listed father, or full sibling) was identifiable since 1977 in the Danish National Patient Register using the *International Classification of Diseases and Related Health Problems*, 8th revision (ICD-8), and 10th revision (ICD-10) codes for multiple sclerosis: ICD-8: 340, ICD-10: G35. Additional outcomes examined were amyotrophic lateral sclerosis, Guillain-Barré syndrome, and myasthenia gravis, all identified in the Danish National Patient Register.

### Covariates

The study included the following potential confounders in the main analysis: (i) the number of live births (categories: 0, 1, 2, and ≥3), (ii) obtainment of a bachelor's degree or higher education (categories: yes, no, or unknown), (iii) family history of multiple sclerosis, defined as

mother, listed father, or full-sibling diagnosed with multiple sclerosis (categories: yes, no, or unknown), (iv) calendar period (categories: 1977–1989, 1990–1999, 2000–2009, and 2010–2017), and (v) maternal age (categories: 12–19, 20–24, 25–29, 30–34, 35–39, 40–44, 45–49, and 50–61 years). Persons with missing data were grouped in a separate 'unknown' category in analyses.

## Statistical analyses

Follow-up was conducted in a time-dependent manner and risk time in years with three decimal places, was split at changes in age, other covariates, or exposure status. The number of events and person-years was summarized, and crude incidence rates per 10,000 person-years were calculated. The primary model estimated crude and adjusted incidence rate ratios (IRR) using Bayesian Poisson regression. A sensitivity analysis used a negative binomial model to better account for potential overdispersion. The Bayesian framework was chosen as it provided an intuitive interpretation of uncertainty in estimates and modeled the complete posterior distribution based on the statistical model, data, and prior beliefs. The chosen priors in the main analysis were modestly informative. A sensitivity analysis used minimally informative uniform priors, approximating estimates which would be obtained by a corresponding frequentist model. The IRR and 95% credible interval (CI) were extracted from the posterior distribution. Convergence of chains was confirmed for all analyses [25]. For detailed specifications of the statistical analyses including chosen priors, see the S2 Appendix.

## Sensitivity analyses

To assess the robustness of the findings, we performed a series of sensitivity analyses. Due to the risk of type I error due to multiple comparisons, these should be interpreted as exploratory. First, as patients diagnosed with multiple sclerosis may have had a prolonged period of prodromal symptoms, a secondary analysis used the date of symptom onset as the event date. The date of symptom of onset was assessed and registered by the clinician upon diagnosing multiple sclerosis. Second, as women who never achieve pregnancy may include a higher proportion of women with underlying medical conditions, these women were excluded. Third, as smoking status (categories: yes/no) is associated with multiple sclerosis, we conducted an analysis including this covariate [26]. Smoking status has in Denmark routinely been assessed for live births and stillbirths since 1997. This analysis only included women with non-missing data in these variables (i.e., complete case analysis) and women were included at the delivery where their smoking status was first registered. The last known values for these variables were used until a newer was available. Fourth, as treatment of pregnancy loss was mainly provided in a hospital setting in Denmark before the year 2000, we conducted an analysis concluding the study this year to minimize exposure misclassification. Fifth, exposure to stillbirth (categories: 0, ≥1) was examined, as stillbirth has been associated with other diseases such as cardiovascular disease [27]. Sixth, an analysis used minimal-informative priors. Finally, we conducted an analysis using a negative binomial model where the covariate calendar year was modeled to have a linear effect, and the covariate age was modeled using a cubic spline with five knots. The knots were chosen as the quartiles of the variable in the full dataset. This analysis improved modelling in case of overdispersion and non-linear effects of age on the outcome.

Analyses were conducted using R version 4.0.3 [28], and Bayesian models were fit using Stan version 2.21.2 [29] using the R interface package rstanarm. A two-sided 95% CI that did not overlap 1.00 was considered statistically significant. The study was reported using the Strengthening the Reporting of Observational Studies in Epidemiology (STROBE) guidelines [30].

## Results

Of the 1,513,560 women eligible for inclusion, 16 were excluded due to a diagnosis of multiple sclerosis before inclusion, resulting in a cohort of 1,513,544 women. The median age at inclusion was 12.0 years (interquartile range [IQR] 12.0–12.2 years), median follow-up time was 28.2 years (IQR 16.0–38.8 years), and the median age at the end of follow-up was 40.4 years (IQR 28.3–50.9 years), with a total of 40,380,194 follow-up years. The cohort development by exposure group and age can be seen in Fig 1. At the end of follow-up, 7,667 women (0.5%) had been diagnosed with multiple sclerosis, the median age at diagnosis 34.2 years (IQR 27.4–41.4 years). At end of follow up, the distribution of persons by pregnancy loss group was: no pregnancy losses: 1,293,791 (85.5%), one loss: 172,150 (11.4%), two losses: 31,757 (2.1%), three or more non-consecutive losses: 4,248 (0.3%), primary recurrent pregnancy loss: 5,635 (0.4%), and secondary recurrent pregnancy loss: 5,963 (0.4%). The number of women diagnosed with multiple sclerosis in each group was: no losses: 6,654 (0.5%), one loss: 807 (0.5%), two losses: 137 (0.4%), three or more non-consecutive losses 12 (0.3%), primary recurrent pregnancy loss: 30 (0.5%), and secondary recurrent pregnancy loss: 27 (0.5%). As seen in Table 1, the adjusted

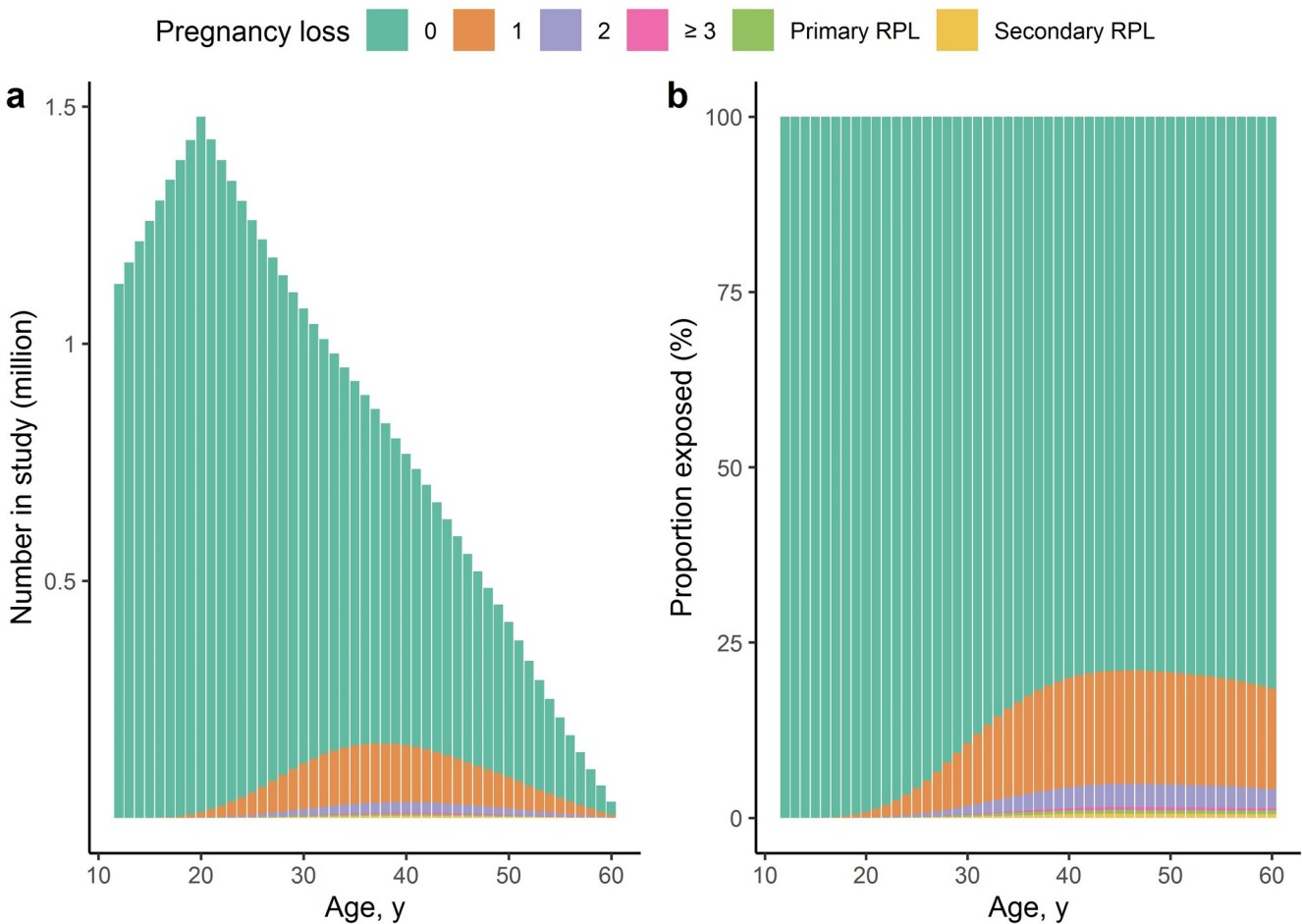

**Fig 1. Cohort development during follow-up.** In a study of the association of pregnancy loss with autoimmune neurological disorder, the entire Danish female population was followed-up in a time-dependent manner from age 12. Women could change exposure status during follow-up in a hierarchal manner from 0 to 1, 2, ≥3 non-consecutive pregnancy losses, primary recurrent pregnancy loss (Primary RPL, i.e. 3 consecutive losses not preceded by a delivery), and secondary recurrent pregnancy loss (Secondary RPL, i.e. 3 consecutive losses preceded by a delivery). The left stacked bar chart (**a**) shows the number at risk by age and exposure group in the cohort. The right chart (**b**) shows the proportion at risk by age and exposure group. The data used to create the figure can be found in the S2 Table.

**Table 1. Association of pregnancy loss with multiple sclerosis, main analysis.**

| Covariate | Events / non-events | Person-years | Incidence rate [a] | Crude IRR [b] (95% CI) | Adjusted IRR [b,c] (95% CI) |
|---|---|---|---|---|---|
| | | | n = 1,513,544 | | |
| Pregnancy loss exposure [d] | | | | | |
| 0 | 6,654 / 1,287,137 | 36,656,369 | 1.82 | 1 | 1 |
| 1 | 807 / 171,343 | 2,986,620 | 2.70 | 1.49 (1.38–1.60) | 1.03 (0.95–1.11) |
| 2 | 137 / 31,620 | 505,490 | 2.71 | 1.48 (1.25–1.75) | 1.02 (0.86–1.20) |
| ≥3 non-consecutive [e] | 12 / 4,236 | 61,249 | 1.96 | 1.06 (0.60–1.73) | 0.81 (0.51–1.24) |
| Primary RPL [f] | 30 / 5,605 | 87,626 | 3.42 | 1.80 (1.24–2.52) | 1.18 (0.84–1.60) |
| Secondary RPL [f] | 27 / 5,936 | 82,839 | 3.26 | 1.71 (1.15–2.44) | 1.16 (0.81–1.63) |
| Number of live births | | | | | |
| 0 | 3,250 / 565,711 | 23,148,201 | 1.40 | 1 | 1 |
| 1 | 1,278 / 242,573 | 5,597,152 | 2.28 | 1.62 (1.52–1.73) | 0.80 (0.75–0.86) |
| 2 | 2,326 / 474,371 | 8,222,068 | 2.83 | 2.01 (1.91–2.12) | 0.86 (0.81–0.92) |
| ≥3 | 813 / 223,222 | 3,412,771 | 2.38 | 1.69 (1.57–1.82) | 0.71 (0.65–0.77) |
| Obtained bachelor's degree | | | | | |
| No | 5,794 / 945,565 | 32,975,926 | 1.76 | 1 | 1 |
| Yes | 1,840 / 504,421 | 6,909,812 | 2.66 | 1.51 (1.44–1.59) | 0.88 (0.83–0.92) |
| Unknown | 33 / 55,891 | 494,455 | 0.67 | 0.40 (0.29–0.54) | 0.90 (0.66–1.20) |
| Family history of MS [g] | | | | | |
| No | 6,906 / 1,341,542 | 37,171,033 | 1.86 | 1 | 1 |
| Yes | 387 / 17,844 | 311,766 | 12.41 | 6.62 (5.97–7.32) | 5.17 (4.65–5.72) |
| Unknown | 374 / 146,491 | 2,897,394 | 1.29 | 0.70 (0.63–0.77) | 0.75 (0.68–0.83) |
| Calendar period | | | | | |
| 1977–1989 | 450 / 33,156 | 7,985,832 | 0.56 | 1 | 1 |
| 1990–1999 | 1,499 / 29,862 | 9,513,828 | 1.58 | 2.74 (2.47–3.05) | 1.56 (1.40–1.74) |
| 2000–2009 | 2,937 / 35,480 | 12,483,749 | 2.35 | 4.10 (3.72–4.52) | 2.08 (1.87–2.31) |
| 2010–2017 | 2,781 / 1,407,379 | 10,396,784 | 2.67 | 4.66 (4.22–5.14) | 2.31 (2.07–2.57) |
| Age group | | | | | |
| 12–19 | 334 / 34,492 | 10,406,128 | 0.32 | 1 | 1 |
| 20–24 | 969 / 217,839 | 6,830,089 | 1.42 | 4.07 (3.62–4.59) | 4.14 (3.66–4.68) |
| 25–29 | 1,350 / 185,044 | 5,819,698 | 2.32 | 6.65 (5.95–7.48) | 6.80 (6.04–7.71) |
| 30–34 | 1,397 / 151,132 | 4,977,287 | 2.81 | 8.05 (7.20–9.04) | 8.08 (7.13–9.19) |
| 35–39 | 1,384 / 151,985 | 4,232,111 | 3.27 | 9.38 (8.38–10.54) | 9.04 (7.95–10.33) |
| 40–44 | 1,096 / 172,693 | 3,415,026 | 3.21 | 9.20 (8.20–10.37) | 8.41 (7.35–9.63) |
| 45–49 | 690 / 180,011 | 2,515,667 | 2.74 | 7.84 (6.93–8.90) | 6.96 (6.04–8.04) |
| 50–61 | 447 / 412,681 | 2,184,188 | 2.05 | 5.84 (5.09–6.71) | 5.10 (4.35–5.94) |

*Abbreviations*: CI, credible interval; IRR, incidence rate ratio; MS, multiple sclerosis

RPL: Recurrent pregnancy loss

[a] Incidence rate per 10,000 person-years.

[b] Estimated using a Poisson model.

[c] Estimates adjusted for the number of live births, obtained bachelor's degree, family history of multiple sclerosis, calendar period, and age group.

[d] Pregnancy loss (i.e. miscarriage) was the exposure of interest defined hierarchically with no pregnancy losses at the lowest level and recurrent pregnancy loss at the highest level.

[e] ≥3 non-consecutive pregnancy losses (not fulfilling criteria for recurrent pregnancy loss).

[f] Recurrent pregnancy loss (RPL) defined as three consecutive pregnancy losses, either preceded by a delivery (secondary) or not (primary).

[g] First degree relative included mother, listed father, or full sibling.

IRR of multiple sclerosis in each group was: one loss: 1.03 (95% CI 0.95–1.11), two losses: 1.02 (95% CI 0.86–1.20), three or more non-consecutive losses: 0.81 (95% CI 0.51–1.24), primary recurrent pregnancy loss: 1.18 (95% CI 0.84–1.60), and secondary recurrent pregnancy loss: 1.16 (95% CI 0.81–1.63), as compared to women with no pregnancy losses. The prior and posterior density distributions of the adjusted IRR can be seen in S1 Fig. One or more live births and obtainment of a bachelor's degree were negatively associated with multiple sclerosis, while calendar year after 1990, age above 20, and family history of multiple sclerosis were positively associated with the multiple sclerosis.

Secondary analyses examined the effect of redefining the outcome date as the date of onset of symptoms (median age at onset 31.2 years, IQR 24.8–38.2), excluding women who never achieved pregnancy, (n = 1,029,795; 68.0% of the cohort), or adjusting for smoking status (n = 793,482; smoking data available for 52.4% of the cohort [among these women 20.1% were registered as ever tobacco smokers]), ending follow-up in the year 2000 (n = 1,107,618; 73.2% of the cohort), using minimally informative priors, or using a negative binomial model with a cubic spline on the covariate age. These secondary analyses did not change the statistical significance of the results, as seen in Table 2.

In the analyses of other autoimmune neurological diseases seen in Table 3, some exposure groups were aggregated due to few events. Adjusted analyses found pregnancy loss or recurrent pregnancy loss was not significantly associated with developing amyotrophic lateral sclerosis, Guillain-Barré syndrome, or myasthenia gravis. During follow-up, seven women with recurrent pregnancy loss developed Guillain Barré syndrome, corresponding to an adjusted IRR of 1.40 (95% CI 0.79–2.37), as compared with women with no pregnancy losses.

## Discussion

This nationwide cohort study, including over 1.5 million women with over 40 million years of follow-up, found no statistically significant association between multiple or recurrent pregnancy loss and later multiple sclerosis, amyotrophic lateral sclerosis, Guillain-Barré syndrome, or myasthenia gravis. The incidence of multiple sclerosis and Guillain-Barré syndrome was estimated to be increased after primary or secondary pregnancy loss as compared to no pregnancy losses, however, the findings were not statistically significant. Considering the rarity of the outcomes in question, an increase in relative risk corresponds to a very low increase in absolute risk.

To our knowledge, no prior studies have examined the association between recurrent pregnancy loss and developing autoimmune neurological disease. A prior register-based cohort study found no association between pregnancy loss and later multiple sclerosis, however exposure to multiple or recurrent pregnancy loss was not investigated [31]. Studies examining the effect of one or more live births on development of multiple sclerosis have shown either no significant effect [32], a decreased risk [31, 33], or a delayed onset of disease [34]. However, the fact that the effect diminishes five years after delivery, and the decrease in risk is also detectable for partners have raised questions of reverse causality. This could for example be caused by the choice to postpone a planned pregnancy during the prodromal stage of multiple sclerosis, consequently an analysis will estimate that nulliparity or low parity to be risk factors for multiple sclerosis [31, 35]. The current study contributes significantly to the current scientific body of evidence examining pregnancy-related factors for autoimmune neurological disorder due to size, length of follow-up, and physician-assigned diagnoses of the primary and additional outcomes.

### Limitations

This study has several limitations. First, as multiple sclerosis is often not diagnosed until years after initial symptoms, the study could have been susceptible to reverse causality. To further

**Table 2. Association of pregnancy loss with multiple sclerosis, sensitivity analyses.**

| Analysis<br>Pregnancy loss exposure [a] | Events | Person-years | Incidence rate [b] | Crude IRR[c] (95% CI) | Adjusted IRR [c, d] (95% CI) |
|---|---|---|---|---|---|
| Symptom onset date as event date | n = 1,513,500 (>99.9%) | | | | |
| 0 | 6,998 | 36,637,219 | 1.91 | 1 | 1 |
| 1 | 743 | 2,984,173 | 2.49 | 1.30 (1.21–1.40) | 1.05 (0.97–1.13) |
| 2 | 114 | 505,044 | 2.26 | 1.18 (0.98–1.40) | 0.97 (0.81–1.15) |
| ≥3 non-consecutive [e] | 10 | 61,228 | 1.63 | 0.87 (0.48–1.46) | 0.82 (0.50–1.28) |
| Primary RPL [f] | 28 | 87,550 | 3.20 | 1.61 (1.10–2.28) | 1.21 (0.85–1.68) |
| Secondary RPL [f] | 21 | 82,781 | 2.54 | 1.29 (0.83–1.90) | 1.08 (0.72–1.54) |
| Excluding never pregnant women | n = 1,029,795 (68.0%) | | | | |
| 0 | 5,128 | 28,608,434 | 1.79 | 1 | 1 |
| 1 | 807 | 2,986,620 | 2.70 | 1.50 (1.39–1.62) | 1.04 (0.97–1.12) |
| 2 | 137 | 505,490 | 2.71 | 1.50 (1.26–1.77) | 1.02 (0.86–1.21) |
| ≥3 non-consecutive [e] | 12 | 61,249 | 1.96 | 1.07 (0.61–1.74) | 0.82 (0.52–1.26) |
| Primary RPL [f] | 30 | 86,642 | 3.46 | 1.84 (1.27–2.58) | 1.22 (0.86–1.67) |
| Secondary RPL[f] | 27 | 82,839 | 3.26 | 1.73 (1.16–2.46) | 1.17 (0.82–1.63) |
| Further adjusted for smoking status | n = 793,482 (52.4%) | | | | |
| 0 | 2,555 | 9,320,898 | 2.74 | 1 | 1 |
| 1 | 564 | 1,953,548 | 2.89 | 1.05 (0.96–1.15) | 1.05 (0.95–1.14) |
| 2 | 99 | 359,427 | 2.75 | 1.00 (0.82–1.22) | 1.00 (0.82–1.22) |
| ≥3 non-consecutive [e] | 10 | 48,229 | 2.07 | 0.78 (0.44–1.32) | 0.77 (0.40–1.33) |
| Primary RPL [f] | 16 | 49,606 | 3.23 | 1.15 (0.69–1.79) | 1.13 (0.68–1.75) |
| Secondary RPL [f] | 20 | 60,095 | 3.33 | 1.19 (0.76–1.76) | 1.18 (0.74–1.77) |
| Ending study in year 2000 | n = 1,107,618 (73.2%) | | | | |
| 0 | 1,667 | 15,679,891 | 1.06 | 1 | 1 |
| 1 | 129 | 651,305 | 1.98 | 1.84 (1.53–2.18) | 1.03 (0.86–1.23) |
| ≥2 non-consecutive [e] | 21 | 93,823 | 2.24 | 1.96 (1.25–2.91) | 1.07 (0.72–1.55) |
| Primary RPL [f] | 5 | 17,997 | 2.78 | 1.93 (0.80–4.02) | 1.29 (0.68–2.30) |
| Secondary RPL [f] | 5 | 12,052 | 4.15 | 2.45 (0.98–5.36) | 1.46 (0.72–2.76) |
| Exposure to stillbirth | n = 1,513,544 (100%) | | | | |
| 0 | 7,638 | 40,254,066 | 1.90 | 1 | 1 |
| ≥1 | 29 | 126,128 | 2.30 | 1.21 (0.83–1.68) | 0.88 (0.64–1.18) |
| Minimally informative priors [g] | n = 1,513,544 (100%) | | | | |
| 0 | 6,654 | 36,656,369 | 1.82 | 1 | 1 |
| 1 | 807 | 2,986,620 | 2.70 | 1.49 (1.38–1.60) | 1.03 (0.95–1.11) |
| 2 | 137 | 505,490 | 2.71 | 1.49 (1.25–1.77) | 1.02 (0.86–1.21) |
| ≥3 non-consecutive [e] | 12 | 61,249 | 1.96 | 1.06 (0.57–1.79) | 0.73 (0.39–1.24) |
| Primary RPL [f] | 30 | 87,626 | 3.42 | 1.87 (1.29–2.64) | 1.22 (0.84–1.71) |
| Secondary RPL [f] | 27 | 82,839 | 3.26 | 1.78 (1.19–2.55) | 1.22 (0.82–1.75) |
| Negative binomial model [h] | n = 1,513,544 (100%) | | | | |
| 0 | 6,654 | 36,656,369 | 1.82 | 1 | 1 |
| 1 | 807 | 2,986,620 | 2.70 | 1.49 (1.38–1.60) | 1.09 (0.98–1.21) |
| 2 | 137 | 505,490 | 2.71 | 1.48 (1.25–1.75) | 1.05 (0.86–1.26) |
| ≥3 non-consecutive [e] | 12 | 61,249 | 1.96 | 1.06 (0.60–1.73) | 0.75 (0.40–1.27) |
| Primary RPL [f] | 30 | 87,626 | 3.42 | 1.80 (1.24–2.52) | 1.26 (0.86–1.80) |

(*Continued*)

**Table 2.** (Continued)

| Analysis | Events | Person-years | Incidence rate [b] | Crude IRR[c] (95% CI) | Adjusted IRR [c, d] (95% CI) |
|---|---|---|---|---|---|
| Pregnancy loss exposure [a] | | | | | |
| Secondary RPL [f] | 27 | 82,839 | 3.26 | 1.71 (1.15–2.44) | 1.24 (0.82–1.80) |

*Abbreviations*: CI, credible interval; IRR, incidence rate ratio

RPL: Recurrent pregnancy loss

[a] Pregnancy loss (i.e. miscarriage) was defined hierarchically with no pregnancy losses at the

lowest level and recurrent pregnancy loss at the highest level.

[b] Incidence rate per 10,000 person-years.

[c] Estimated using a Poisson model unless stated otherwise.

[d] Estimates adjusted for the number of live births, obtained bachelor's degree, family history of multiple sclerosis, calendar period, and age group unless otherwise

stated.

[e] ≥3 non-consecutive pregnancy losses (not fulfilling criteria for recurrent pregnancy loss).

[f] Recurrent pregnancy loss (RPL) defined as three consecutive pregnancy losses, either preceded by a delivery (secondary) or not (primary).

[g] Uniform priors for parameters and intercept. *(continued)*

[h] Using a negative binomial model and fitting the covariate calendar year as a linear predictor and the covariate age using a cubic spline with five knots.

explore this, a sensitivity analysis was conducted using the date of symptom onset as the event date. This did not change the findings materially.

Second, women who never achieve pregnancy may comprise a heterogenous group not necessarily comparable with other women. To increase the homogeneity of the comparator

**Table 3. Association of pregnancy loss with other autoimmune neurological disorders.**

| Analysis | Events / non-events | Person-years | Incidence rate [b] | Crude IRR [c] (95% CI) | Adjusted IRR [c, d] (95% CI) |
|---|---|---|---|---|---|
| Pregnancy loss exposure [a] | | | | | |
| Amyotrophic lateral sclerosis | n = 1,513,557 | | | | |
| 0 | 164 / 1,293,328 | 36,731,181 | 0.04 | 1 | 1 |
| 1 | 27 / 172,367 | 2,997,495 | 0.09 | 1.92 (1.26–2.79) | 1.08 (0.75–1.51) |
| ≥2 or RPL[e,f] | 5 / 47,666 | 739,973 | 0.07 | 1.40 (0.62–2.78) | 0.95 (0.57–1.54) |
| Guillain-Barré syndrome | n = 1,513,492 | | | | |
| 0 | 630 / 1,292,860 | 36,721,402 | 0.17 | 1 | 1 |
| 1 | 62 / 172,286 | 2,996,260 | 0.21 | 1.19 (0.92–1.54) | 1.12 (0.86–1.43) |
| ≥2 (not including RPL) [e] | 13 / 36,031 | 568,465 | 0.23 | 1.28 (0.73–2.07) | 1.18 (0.75–1.79) |
| RPL [e,f] | 7 / 11,603 | 171,149 | 0.41 | 1.95 (0.92–3.70) | 1.40 (0.79–2.37) |
| Myasthenia gravis | n = 1,513,528 | | | | |
| 0 | 404 / 1,293,099 | 36,726,232 | 0.11 | 1 | 1 |
| 1 | 53 / 172,314 | 2,996,773 | 0.18 | 1.58 (1.18–2.08) | 1.20 (0.92–1.57) |
| ≥2 or RPL [e,f] | 6 / 47,652 | 739,746 | 0.08 | 0.83 (0.40–1.52) | 0.85 (0.54–1.31) |

*Abbreviations*: CI, credible interval; IRR, incidence rate ratio

RPL: Recurrent pregnancy loss

[a] Pregnancy loss (i.e. miscarriage) was the exposure of interest defined hierarchically with no pregnancy losses

at the lowest level and recurrent pregnancy loss at the highest level.

[b] Incidence rate per 10,000 person-years

[c] Estimated using a Poisson model

[d] Estimates adjusted for the number of live births, obtained bachelor's degree, family history of the outcome of interest, calendar period, and age group.

[e] Groups aggregated due to few events.

[f] Recurrent pregnancy loss (RPL) defined as three consecutive pregnancy losses, either preceded by a delivery (secondary) or not (primary).

group, women who were never registered pregnant were excluded in a sensitivity analysis. Again, this did not change the significance of the results. Third, although the primary analysis adjusted for many important confounders, residual confounding could not be ruled out. Therefore, an analysis further adjusted for smoking status as a proxy for unmeasured lifestyle factors, and the resulting estimates were materially unchanged. Smoking status was only known from year 1997 an onwards for women with a delivery (ie. live- or stillbirth), and consequently cannot necessarily be extrapolated to women with no prior deliveries. Therefore, a post-hoc sensitivity analysis (S3 Table), explored the effect of identifying smokers by hospital diagnosis code (ICD-10: F17) or fulfilling a prescription for a smoking cessation drug (Anatomical Therapeutic Chemical Classification code: N07BA), this did not change the significance of the results. Fourth, as the exposure only included clinical pregnancy losses, that is, those treated in a hospital setting and not including those only treated by private practitioners, misclassification of the exposure may have biased results. Assuming non-differential misclassification, this would bias our results towards the null. Until the year 2000, pregnancy losses in Denmark were routinely evaluated in a hospital setting [9]; therefore, a secondary analysis evaluated the effect of ending the study on December 31, 1999. Although the statistical power was reduced, this did not change the significance of the results. Further, a study found that a diagnosis of miscarriage in the National Patient Register had a high validity as the diagnosis was confirmed in 114 out of 117 hospital records [36].

Fifth, conducting Bayesian models facilitates and necessitates incorporating prior evidence and knowledge into the probability of an outcome. However, in the current study, only sparse evidence existed to guide the model, and the prior was used to guide posterior estimates into a plausible range. To further investigate the effect of this, a secondary model assessed the effect of minimally informative uniform priors on estimates. This analysis only changed the results minimally due to the abundant data guiding the model. Fifth, a sensitivity analysis was designed to better handle potential overdispersion and non-linear effects of the covariate age. This analysis did not change results substantially. Misclassification of the outcome was assumed to be minimal due to the high completeness (91%) and validity (94%) of the Danish Multiple Sclerosis Registry. Additional outcomes were based on diagnosis codes in the Danish Patient Register which have shown positive predictive values ranging from 83.8 to 92.5% [24, 37–39]. Family linkage was based on the Civil Registration System and has to our knowledge not been validated. Correct assignment of the mothers is likely very accurate as is occurs right after birth. Family history of multiple sclerosis was based on diagnosis codes in the Danish National Patient Register which have been validated against the Danish Multiple Sclerosis Registry and shown a completeness of 92.8% and validity of 95.1% [40]. In the cohort missing values were rare: 9.7% had missing information on one or both parents leading to unknown family history of an outcome, and 3.7% had missing information on educational status. We acknowledge this could be a potential source of bias.

Although the population was large, some exposure groups and outcomes were rare. Therefore, the possibility of insufficient power to accurately detect a small increase in risk cannot be excluded, and future studies may well aim to reproduce estimates for the outcomes of multiple sclerosis and Guillain-Barré syndrome after exposure to recurrent pregnancy loss.

## Conclusion

This nationwide study found no significant association between pregnancy loss and multiple sclerosis, amyotrophic lateral sclerosis, Guillain-Barré syndrome, or myasthenia gravis. This evidence should be reassuring to women already burdened by the loss of one or more pregnancies.

## Supporting information

**S1 Fig. Prior and posterior probability density.**
(DOCX)

**S1 Table. Registers and definitions used.**
(DOCX)

**S2 Table. Data used to create Fig 1.**
(DOCX)

**S3 Table. Association of pregnancy loss with multiple sclerosis, further adjusted for smoking status defined by hospital admission for smoking (ICD-10: F17) or fulfilling prescription for a smoking cessation drug (ATC: N07AB).**
(DOCX)

**S1 Appendix. Restriction periods between pregnancies.**
(DOCX)

**S2 Appendix. Specification of the Bayesian model.**
(DOCX)

## Author Contributions

**Conceptualization:** Anders Pretzmann Mikkelsen, Pia Egerup, Astrid Marie Kolte, David Westergaard, Henriette Svarre Nielsen, Øjvind Lidegaard.

**Data curation:** Anders Pretzmann Mikkelsen, Øjvind Lidegaard.

**Formal analysis:** Anders Pretzmann Mikkelsen.

**Funding acquisition:** Anders Pretzmann Mikkelsen, Henriette Svarre Nielsen.

**Investigation:** Anders Pretzmann Mikkelsen.

**Methodology:** Anders Pretzmann Mikkelsen, David Westergaard.

**Project administration:** Anders Pretzmann Mikkelsen.

**Resources:** Anders Pretzmann Mikkelsen.

**Software:** Anders Pretzmann Mikkelsen.

**Supervision:** Øjvind Lidegaard.

**Validation:** Anders Pretzmann Mikkelsen.

**Visualization:** Anders Pretzmann Mikkelsen.

**Writing – original draft:** Anders Pretzmann Mikkelsen.

**Writing – review & editing:** Anders Pretzmann Mikkelsen, Pia Egerup, Astrid Marie Kolte, David Westergaard, Henriette Svarre Nielsen, Øjvind Lidegaard.

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
