## [Decision Letter · Decision Letter 0]

11 Feb 2022

PONE-D-21-39062Pregnancy loss and risk of multiple sclerosis and autoimmune neurological disorder:  A nationwide cohort studyPLOS ONE

Dear Dr. Mikkelsen,

Thank you for submitting your manuscript to PLOS ONE. After careful consideration, we feel that it has merit but does not fully meet PLOS ONE’s publication criteria as it currently stands. Therefore, we invite you to submit a revised version of the manuscript that addresses the points raised during the review process.

We look forward to receiving your revised manuscript.

Kind regards,

Angela Lupattelli, PhD

Academic Editor

PLOS ONE

Journal Requirements:

(This work was supported by The Research Fund of Rigshospitalet, Copenhagen University Hospital [grant number E-22515-01] awarded to APM and ØL. URL: https://www.forskningspuljer-rh.dk/

Ole Kirks Foundation [no grant number] awarded to HSN. URL: https://www.olekirksfond.dk/)

(I have read the journal's policy and the authors of this manuscript have the following competing interests: The authors report no conflicts of interest for the submitted research. HSN reports grants from Freya Biosciences ApS, Ferring Pharmaceuticals, BioInnovation Institute, Ministry of Education, Novo Nordisk Foundation, Augustinus Foundation, Oda and Hans Svenningsens Foundation and honoraria from Ferring Pharmaceuticals, Merck A/S, Astra Zenica, Cook Medical, and DW reports a grant from Novo Nordisk Foundation during the conduct of this study.)

We note that you received funding from a commercial source: (Freya Biosciences ApS, Ferring Pharmaceuticals, BioInnovation Institute, Ministry of Education, Novo Nordisk Foundation, Augustinus Foundation, Oda and Hans Svenningsens Foundation, Merck A/S, Astra Zenica, Cook Medical,Novo Nordisk)

Reviewers' comments:

Reviewer's Responses to Questions

**Comments to the Author**

1. Is the manuscript technically sound, and do the data support the conclusions?

Reviewer #1: Yes

Reviewer #2: Yes

2. Has the statistical analysis been performed appropriately and rigorously? 

Reviewer #1: Yes

Reviewer #2: Yes

3. Have the authors made all data underlying the findings in their manuscript fully available?

Reviewer #1: Yes

Reviewer #2: Yes

4. Is the manuscript presented in an intelligible fashion and written in standard English?

Reviewer #1: Yes

Reviewer #2: Yes

5. Review Comments to the Author

Reviewer #1: Well done strudy with rigorous analysis. Excellent that researchers looked at symptom onset rather than date of diagnosis for MS onset. In general therwe was little pregnancy loss. Surprislingly with 52.4% of the cohort smokers, smoking had litle impact on pregnancy loss or risl for MS. The authors explained this well. The appendix contsained a detailed review ofthe Bayesian Model- appreciated.

Although these questions will not hold publication, I wondered what was the definition of a "sensitgivity analysis", when was the study completed? and definitions for the nongynecologist would be helpful (blighteed ovum? and missed abortion?)

Well done- this may go to press

Reviewer #2: This is well written paper from Denmark utilizing the Danish nationwide register data. Authors studied whether pregnancy loss (miscarriage) is associated with autoimmune neurological disorders including multiple sclerosis (MS), amyotrophic lateral sclerosis, Guillain-Barré syndrome and myasthenia gravis. Study material consisted of female population aged 12 and older living in Denmark between 1977 and 2017. Findings from the main analysis and bunch of secondary and sensitivity analyses were reassuring: pregnancy loss was not significantly associated with autoimmune neurological disorder. Previous live births and higher education were negatively associated with MS while calendar year, age and family history increased the risk of MS. Conclusions are supported by the data.

Study exposure i.e. pregnancy loss was identified from the Danish Patient Register using diagnostic codes that had been previously validated. Also, an algorithm to identify same and different pregnancy episode are clinically justified. Authors write that they may not have captured early spontaneous pregnancy losses that does not require treatment at the hospitals. There is no reason to anticipate misclassification according to disease status. Also, sensitivity analysis was conducted restricting the study end into year 2000 as before that miscarriages were mainly treated at hospitals.

Study outcome was incident diagnosis recorded at the Danish Multiple Sclerosis Register with high completeness and validity. Secondary outcomes were identified from the Danish Patient Register.

Authors have conducted Bayesian Poisson regression to estimate adjusted incidence rate ratios and 95% credible intervals. Bayesian approach is justified based on rare event. However, Bayesian approach necessitates prior information on the probability of the outcome, but only sparse data existed to guide the model. A sensitivity analysis using a negative binomial model was done which did not change results substantially.

The study presents the results of the original research and have not been published elsewhere. Study material and analyses are described in sufficient detail and study reports scientifically sound results following the STROBE guidelines. Data cannot be made available due to Danish law for secondary use of register data. I have mainly minor issues where text could be improved and/or clarification added.

On p. 6 authors describe smoking as a proxy for unmeasured lifestyle and environmental factors possibly associated with outcome. Only women with non-missing data were included in the sensitivity analysis and sample size was limited almost to a half. Also, smoking status was only known for women after a delivery and cannot be extrapolated to women without delivery history. How this could affect the results? Would there be a possibility to identify smokers from the patient registry data e.g. ICD-10 F17? Or drug refills of varenicline ATC code N07A03? The missing information on smoking in pregnancies ending before viability seems a limitation that has not been fully addressed. Possibly associated with the outcome is an understatement as in the literature smoking has been associated both with increased risk of disease and with MS disease activity (see eg. https://pubmed.ncbi.nlm.nih.gov/32259516/).

Missing information is always a challenge in register-based studies. Complete-case analysis would restrict the sample size substantially. Modelling missing data as a separate category is an alternative, however that may result in study findings that are hard to interpret. Have you considered multiple imputation? In Table 1, unknown information on education was significantly associated with MS. Any ideas what could explain the finding? It is also uncertain where does unknown family history derive from? First degree relatives not found from the Patient Registry? Look-back period can have an impact on how reliable information can be obtained for the parents. Please explain and add into p.5 line 108 what years were available from the Danish Patient registry to identify MS in first degree relatives.

-abstract says age 11 while on page 4 age 12

-p3. lines 59-64 extremely long and fluctuating sentence. Please split. Also, the concept of secondary and primary pregnancy loss needs to be explained here when mentioned for the first time in the main text.

-p4 line 90 other pregnancies were identified but not directly used in analyses. What do you mean? Was used or wasn´t used?

-p5 line 101 3 or more pregnancy losses (not fulfilling criteria for recurrent pregnancy loss) is a bit unclear. It should be possible to understand the design without S1 Table. It is crucial to understand the difference between category 3+ and recurrent primary and secondary as these are repeated in result tables and text with often slightly different working in the brackets. Couldn´t you just simply name the category ≥ 3 non-consecutive pregnancy losses?

-p5 line 105 explain the difference between date of diagnosis and onset of disease? Which one was used as date of incidence in the analysis? In p. 6 sensitivity analysis date of symptom onset was used as an event date. Is onset of disease the same as date of symptom onset? Check the wording and be consistent.

p.6 line 140 delete of

p.7 line 151. Rationale for studying exposure to stillbirth is lacking. Also, the definition changed in the Danish Medical Birth Register over time from 28 weeks before 2004 to 22 weeks, how that was taken into account in the analysis?

p.14 lines 261-263 data quality for secondary outcomes in Danish Patient Register is provided. What about the validity of MS diagnosis i.e. capture of familial disease which was adjusted in the models?

S1 Figure. Should the red distribution be 0.25-4.00 in the figure? Mean being IRR=1 upper credible limit cannot be 0.40, right?

6. PLOS authors have the option to publish the peer review history of their article (what does this mean?). If published, this will include your full peer review and any attached files.

Reviewer #1: **Yes: **Heidi Maloni PhD

Reviewer #2: **Yes: **Maarit Leinonen

---

## [Author Response · Author response to Decision Letter 0]

4 Mar 2022

Please see the attached file named "Response to Reviewers.docx"

---

## [Editor Report · Decision Letter 1]

16 Mar 2022

Pregnancy loss and risk of multiple sclerosis and autoimmune neurological disorder:  A nationwide cohort study

PONE-D-21-39062R1

Dear Dr. Mikkelsen,

We’re pleased to inform you that your manuscript has been judged scientifically suitable for publication and will be formally accepted for publication once it meets all outstanding technical requirements.

Kind regards,

Angela Lupattelli, PhD

Academic Editor

PLOS ONE

---

## [Editor Report · Acceptance letter]

22 Mar 2022

PONE-D-21-39062R1 

Pregnancy loss and risk of multiple sclerosis and autoimmune neurological disorder:  A nationwide cohort study 

Dear Dr. Mikkelsen:

I'm pleased to inform you that your manuscript has been deemed suitable for publication in PLOS ONE. Congratulations! Your manuscript is now with our production department. 

Kind regards, 

on behalf of

Dr. Angela Lupattelli 

Academic Editor

PLOS ONE